# 12/111phiA Prophage Domestication Is Associated with Autoaggregation and Increased Ability to Produce Biofilm in *Streptococcus agalactiae*

**DOI:** 10.3390/microorganisms9061112

**Published:** 2021-05-21

**Authors:** Adélaïde Renard, Seydina M. Diene, Luka Courtier-Martinez, Julien Burlaud Gaillard, Houssein Gbaguidi-Haore, Laurent Mereghetti, Roland Quentin, Patrice Francois, Nathalie Van Der Mee-Marquet

**Affiliations:** 1UMR INRAE 1282 Infectiologie et Santé Publique, Bactéries et Risque Materno-Foetal, Université de Tours, 37000 Tours, France; adelaide.renard@etu.univ-tours.fr (A.R.); n.brion@chu-tours.fr (L.C.-M.); mereghetti@univ-tours.fr (L.M.); rlq37000@gmail.com (R.Q.); 2Aix-Marseille Université, MEPHI, IRD, APHM, IHU-Méditerranée Infection, Faculté de Pharmacie, 13000 Marseille, France; seydina.m.ddiene@gmail.com; 3Plateforme IBiSA Microscopie Electronique, Faculté de Médecine, Université de Tours, 37000 Tours, France; julien.gaillard@univ-tours.fr; 4Service d’Hygiène Hospitalière, CHRU, 25056 Besançon, France; hhgbaguidihaore@chu-besancon.fr; 5Laboratoire de Recherche Génomique, Service des Maladies Infectieuses, Centre Médical Universitaire, Hôpitaux Universitaire de Genève, 1205 Geneva, Switzerland; patrice.francois@genomic.ch

**Keywords:** *Streptococcus agalactiae*, phage, autoaggregation, biofilm, pathogenicity, neonate

## Abstract

CC17 *Streptococcus agalactiae* carrying group-A prophages is increasingly responsible for neonatal infections. To investigate the impact of the genetic features of a group-A prophage, we first conducted an in silico analysis of the genome of 12/111phiA, a group-A prophage carried by a strain responsible for a bloodstream infection in a parturient. This revealed a Restriction Modification system, suggesting a prophage maintenance strategy and five ORFs of interest for the host and encoding a type II toxin antitoxin system RelB/YafQ, an endonuclease, an S-adenosylmethionine synthetase MetK, and an StrP-like adhesin. Using the WT strain cured from 12/111phiA and constructing deleted mutants for the ORFs of interest, and their complemented mutants, we demonstrated an impact of prophage features on growth characteristics, cell morphology and biofilm formation. Our findings argue in favor of 12/111phiA domestication by the host and a role of prophage features in cell autoaggregation, glycocalyx and biofilm formation. We suggest that lysogeny may promote GBS adaptation to the acid environment of the vagina, consequently colonizing and infecting neonates.

## 1. Introduction

*Streptococcus agalactiae* or Group B Streptococcus (GBS) is a leading bacterial pathogen responsible for neonatal infection [1]. GBS strains belonging to the hypervirulent clonal complex 17 (CC17) are responsible for invasive infections associated with high rates of morbidity and mortality in neonates [2,3]. Since two decades ago, following the introduction of antimicrobial prophylaxis at the time of delivery in parturients showing vaginal colonization with GBS, the incidence of early onset disease (i.e., infections occurring during the first week of life) has significantly decreased [4,5,6,7]. This decreasing trend is not, however, observed with late-onset GBS disease (LOD) (i.e., infections occurring in neonates between 7 and 90 days old) [4,5]. The factors driving the pathogenicity of GBS and contributing to the increasing incidence in neonatal disease for the past two decades remain poorly understood.

Temperate bacteriophages may integrate their genome into the bacterial chromosome (prophages) and confer evolutionary or adaptive advantage to the host [8]. The acquisition of the phiUK-M3.1 prophage by *Streptococcus pyogenes* has been associated with the emergence of a lineage named emm/M3, showing a high ability to infect humans [9]. In *Staphylococcus aureus*, the acquisition of prophage elements has been associated with bacterial host jumps [10,11,12]. Prophage features may result in increasing bacterial fitness or virulence [8]. The expansion of an emm12 *S. pyogenes* clone has been associated with the acquisition of phiHKU.ssa and phiHKU.vir, two prophages carrying genes encoding SSA, Spe and Spd1 exotoxins [13]. In *S. aureus* species, phiPVL-like prophages have been shown to carry genes encoding Panton–Valentine leukocidin (PVL) [12,14], and the acquisition of an Sa3 prophage by swine-associated *S. aureus* belonging to clonal complex 398 has been associated with the emergence of a human-adapted lineage with increasing bacterial adhesion capacity [15].

Using whole-genome sequencing (WGS) of strains representative of the species, we previously characterized six genetic groups of GBS prophages (A to F) [16]. The study of 123 GBS strains isolated over the period 2016–2018, from infected Czech patients and colonized parturients, revealed that 71% of isolates carry group-A prophages [17]. Relying on the prophage content of 106 GBS strains responsible for infection in French neonates, we demonstrated that CC17 isolates carrying group-A prophages have been increasingly responsible for LOD over the past two decades [18]. WGS of 1345 GBS isolated from infected Dutch neonates (1987–2016) recently revealed a new lineage (CC17-1A), whose strains mostly harbor a prophage named phiStag1, carrying a highly conserved *strP* gene encoding a protein highly suggestive of an adhesion protein StrP [19].

The aim of our study was to investigate the impact of genetic features of a group-A prophage, 12/111phiA, on a CC17 strain responsible for a bloodstream infection in a parturient. For this purpose, we first conducted an in silico analysis of the 12/111phiA genome to identify ORFs that could be potentially involved in the increasing fitness and virulence of the recipient host GBS. We cured the prophage from the WT strain, constructed deleted mutants for the ORFs of interest, and constructed their respective in situ and plasmid-complemented mutants. We then compared the phenotype of the WT strain to that of the constructions for their respective growth characteristics, morphology via optical, scanning, and transmission electron microscopy, and their ability to produce biofilm.

## 2. Materials and Methods

### 2.1. Bacterial Strains and Culture Conditions

The GBS and *E. coli* strains used in the study are listed in Appendix A. Todd-Hewitt (TH) (Sigma-Aldrich, Saint-Louis, MO, USA) agar and broth medium, Man Rogosa Sharpe (MRS) (VWR, Leuven, Belgium) broth medium, TH broth medium supplemented with 1% glucose (THB 1% gl), modified TH broth (mTH) [20] and 5% horse blood Trypticase soy agar plates (1.5% agar; bioMérieux, Marcy-l’Etoile, France) were used to cultivate GBS. GBS harbouring plasmid pG+host1^TS^ or pTCV-P_Tet_ were sustained in medium containing 10 µg/mL erythromycin. *E. coli* XLI strain was used for pG+host1^TS^ recombinant plasmid and pTCV-P_Tet_ complementation plasmid selection (Appendix A). Luria-Bertani (LB, MP, Solon, OH, USA) agar plates or liquid medium were used to cultivate *E. coli* strains at 37 °C with agitation (200× rpm). *E. coli* harbouring plasmid pG+host1^TS^ and pTCV-P_Tet_ were sustained in media containing 150 µg/mL erythromycin and 25 µg/mL Kanamycin, respectively.

### 2.2. Construction of 12/111∆phiA

The 12/111ΔphiA prophage-free strain was obtained from the wildtype (WT) 12/111 strain following exposure to 1 µg/mL mitomycin C for 2 h, using a protocol described by Domelier et al. [20]. After exposure to mitomycin C, the bacterial pellet was resuspended in 1 mL mTH, diluted to 10^−7^ and plated on a TSH agar plate. After incubation at 37 °C for 18 h, colonies were screened for prophage loss by PCR [16,18]. Whole-genome sequencing of the 12/111ΔphiA strain and pairwise comparison with the WT 12/111 strain were performed to check the absence of prophage, the closure of the insertion site where 12/11phiA was initially inserted, and the absence of secondary mutation. We searched for phage production following exposure of WT 12/111 to mitomycin C. For this purpose, after exposure to mitomycin C, the bacterial culture was centrifuged (1500× *g*; 15 min) and the supernatant filtered using a 0.45 µm-pore-size filter (Millipore, Bedford, MA, USA), treated with DNase (TurboTM DNase, Ambion, Austin, TX, USA) and tested for phage by PCR [16].

### 2.3. 12/111phiA Genome Analysis

Prophage Hunter software [21] was used to extract the 12/111phiA genome from the whole-genome sequence of WT 12/111 strain [16] and to annotate the function of phage proteins. Alignments of prophage proteins with functionals bacterial homologs were performed using EMBL-EBI software [22]. The modular structure of 12/111phiA was determined by analogy with Sfi21 organization [23,24,25].

### 2.4. Construction of Deletion Mutants

The mutants were obtained from the WT 12/111 strain. The nonpolar deletion mutant 12/111Δ*relB-metK* was obtained by double homologous recombination of the region starting with the *relB* coding sequence and ending with the *metK* coding sequence. PCR of flanking regions of the *relB-metK* region was performed using primer pairs AR1/AR2 and AR3/AR4. A recombination cassette was constructed, consisting in the combination of the previous two amplifications by splicing-by-overlap extension PCR with primers AR1 and AR4 (Appendix A). The fragment obtained was cloned into the BamHI/EcoRI restriction sites of the pG+host1 plasmid (Appendix A). The mutants 12/111∆*relB-yafQ*, 12/111∆*endonuclease-metK*, 12/111∆*endonuclease* and 12/111∆*metK* were constructed according to the same protocol. The recombinant plasmids were transformed by electroporation into *E. coli* XL1 for amplification and purification as previously described [26,27] using the Micropulser (Bio-Rad, Hercules, CA, USA) and Ec2 conditions (2.5 kV) with 1 to 2 μg of the appropriate plasmids. The transformants were selected on LB agar with the appropriate antibiotics. After plasmid verification by sequencing, plasmids were transformed into 12/111 GBS strain; transformants were selected on 5% horse TSH agar plates (bioMérieux, Marcy-l’Etoile France). Allelic exchange was performed as previously described [28]. Deletions of the region of interest were checked by PCR and sequencing. Standard OneTaq polymerase (New England BioLabs (NEB), Ipswich, MA, USA) was used for verification PCR, Q5 high-fidelity DNA polymerase (NEB) was used for cloning PCR and sequencing reaction. For sequencing, PCR fragments were purified with NucleoSpin gel and PCR cleanup NucleoSEQ kit (Macherey-Nagel, Düren, Germany), according to the manufacturer’s instructions. Sequencing of both strands was performed by using the BigDye Terminator v. 3.1 cycle sequencing kit (Applied Biosystems, Foster City, CA, USA) and the ABI Prism 310 genetic analyser (Applied Biosystems, Foster City, CA, USA).

### 2.5. Construction of In Situ Complemented Mutants

In situ complemented mutants were obtained using pG+host1. 12/111Δ*relB-metK::relB-metK* was obtained by amplification of a region overlapping the flanking region of *relB-metK* with primer pairs AR1/AR4. The same cloning method was used to obtain 12/111∆*relB-yafQ::relB-yafQ*, 12/111∆*endonuclease-metK:endonuclease-metK*, 12/111∆*endonuclease::endonuclease* and 12/111∆*metK::metK*. Complementation inserts of *relB-yafQ*, e*ndonuclease-metK*, *endonuclease* and *metK* were amplified with primer pairs described in Appendix A. The fragments obtained were cloned into the BamHI/EcoRI restriction sites of pG+host1, before electroporation first into *E. coli* XL1, then into appropriate deleted mutants after plasmid sequencing. Allelic exchange was performed as previously described [28]. Complementation of regions/ORFs was checked by PCR and sequencing.

### 2.6. Construction of Plasmid-Complemented Mutants

To complement the 12/111Δ*relB-metK* mutant and 12/111ΔphiA strain with a plasmid by *relB-yafQ* region*,* a region starting with *relB* and ending with *yafQ* was amplified by PCR, using primer pairs AR33/AR34. Using BamHI/PstI restriction sites, amplification products were cloned into pTCV-P_Tet_ [29] (Appendix A). Using the same strategy, the 12/111Δ*relB-metK* and 12/111ΔphiA mutant strains were complemented with the entire sequence of the *endonuclease-metK* region, *endonuclease* and *metK* using the primers described in Appendix A. Primers AR34, AR35 and AR85 yielded an extension with a transcriptional terminator sequence ensuring transcriptional end of cloning insert. The recombined pTCV-P_Tet_ plasmids were electroporated into *E. coli* XLI cells for amplification and purification and into the 12/111 mutants after plasmid verification.

The set of 35 constructs was stored at −80 °C in TH broth supplemented with 20% glycerol.

### 2.7. Whole-Genome Sequencing

High-throughput sequencing technology was used to sequence the GBS genomes. Genomic DNA was purified using a DNeasy kit (Qiagen, Courtaboeuf, France) per the manufacturer’s instructions, including RNAse A treatment. Purity and quality were assessed using NanoDrop and Qubit (ThermoFisher Scientific, Waltham, MA, USA). Fragmentation was performed using 1 μg of purified genomic DNA and the TruSeq DNA LT Sample Preparation Kit according to the manufacturer’s protocol (Illumina, San Diego, CA, USA; Cat. No. FC-121-2001). The resulting libraries were pooled into a single library for paired-end sequencing of 2 × 100-bp on the Illumina HiSeq 2500 using the TruSeq PE Cluster Kit v. 3. Data were processed using the Illumina Pipeline Software package and aligned using Eland v. 2e. Whole-genome sequences of the strains sequenced in this study have been deposited in the NCBI database under bioproject number. Genome sequence accession numbers are as follow: 12/111 SAMN06329889, 12/111ΔphiA JAGMTM000000000, 12/111Δ*relB-metK* n°1 JAGKKV000000000, 12/111Δ*relB-metK::relB-metK* n°1 JAGKKW000000000, 12/111Δ*relB-metK* n°2 JAGKKX000000000 and 12/111Δ*relB-metK::relB-metK* n°2 JAGKKY000000000.

### 2.8. Bioinformatics Analyses

Read sequence quality was assessed using the Fastqc program (http://www.bioinformatics.babraham.ac.uk/projects/fastqc, accessed on 17 January 2021). Reads were quality-filtered with fastq-mcf (Ea-utils: https://expressionanalysis.github.io/ea-utils, accessed on 17 January 2021). Genome assembly was performed with the Edena v. 3 assembler [30]. The assembled genomes were annotated using the PATRIC server (https://www.patricbrc.org, accessed on 20 January 2021). Pairwise comparison of genome sequences was performed to identify non-synonymous mutations from the analyzed mutants and their complements. The core proteome of each strain was identified and the pairwise average similarity was determined. All detected non-synonymous mutations between strains were investigated and confirmed by mapping reads against the reference gene of interest using CLC genomics software. Comparison of prophage regions between strains was performed using EasyFig software [31].

### 2.9. Growth Kinetics

From an overnight culture of GBS in THB, THB/MRS (50:50) was inoculated (optical density at 600 nm (OD600), 0.05) in triplicate in 96-well microtiter plates (flat bottom; Evergreen) (300 µL culture volume). The plates were incubated for 24 h at 37 °C with continuous shaking in an Eon thermo-regulated spectrophotometer plate reader (BioTek Instruments, Winooski, VT, USA). OD600 was measured every 20 min. Each manipulation was performed in triplicate. For each strain, the average OD600 at 23 h culture determined the maximum growth value. After 24 h in the microplates, cell counts were performed after dilution (10^−1^ to 10^−4^) and depositing 100 µL of each dilution onto TSH. Colony forming units were counted after incubation overnight at 37 °C.

### 2.10. Microscopic Analysis

#### 2.10.1. Gram Staining

GBS strains were inoculated (OD600, 0.05) in THB 1% glucose medium in a tube from an overnight culture in the same medium. After static incubation at 37 °C for 24 h, Gram staining was performed [32,33].

#### 2.10.2. Scanning Electron Microscopy (SEM)

GBS strains were cultured in THB 1% gl for 18 h. From this culture, GBS strains were seeded in the same medium, to obtain 10 mL culture with (OD600, 0.05), and incubated at 37 °C for 24 h. The broths were then centrifuged at 1500× rpm; the bacterial cells were washed twice with phosphate-buffered saline (PBS), and fixed for 24 h in 4% paraformaldehyde, 1% glutaraldehyde (Sigma, St-Louis, MO, USA) in 0.1 M phosphate buffer (pH 7.2). A second fixation was performed after washing in PBS using 2% osmium tetroxide (Agar Scientific, Stansted, UK) for 1 hr. Cells were dehydrated in a graded series of ethanol solutions and dried in hexamethyldisilazane (HMDS, Sigma, USA). Samples were coated with 40 Å platinum, using a GATAN PECS 682 device (Pleasanton, CA, USA), before observation under a Zeiss Ultra plus FEG-SEM scanning electron microscope (Oberkochen, Germany) [34]. The GBS strains were studied for cell clustering and morphology.

#### 2.10.3. Transmission Electron Microscopy (TEM)

GBS strains were prepared as for SEM. Cells were dehydrated in a graded series of ethanol solutions and propylene oxide. Impregnation was performed with a mixture of (1:1) propylene oxide/Epon resin (Sigma, USA) and left overnight in pure resin. Cells were embedded in Epon resin (Sigma, USA) for 48 h at 60 °C. Ultra-thin sections (90 nm) of these blocks were obtained with a Leica EM UC7 ultramicrotome (Wetzlar, Germany). Sections were stained with 5% uranyl acetate (Agar Scientific, UK), 5% lead citrate (Sigma, USA) and observations were made with a transmission electron microscope (JEOL 1011, Tokyo, Japan). Wall thickness was measured using ImageJ software [35]. The frequency of cocci yielding glycocalyx-like structures was evaluated as the percentage of cells with glycocalyx among 200 cells counted per strain.

### 2.11. Biofilm Formation Assay

Biofilm formation was studied using polystyrene 96-well microplates (Costar, Washington, DC, USA) and a modified protocol [36,37]. From overnight cultures in THB supplemented with 1% glucose (three independent cultures per strain), a dilution in the same medium was performed to obtain an OD600 of 0.05. Then, 200 µL of this dilution was dispensed in quadruplicate into the microplate and incubated at 37 °C for 48 h. After incubation, the culture was removed by inverting the plate and shaking, washing twice by gentle submersion in water, and shaking out the water. Biofilms were strained with 0.2% crystal violet (Merck, Darmstadt, Germany) for 15 min (110 µL per well), washed four times with water, and air-dried. Biofilm formation was quantified by dissolution of biofilm in ethanol/acetone (80:20) after 2 h at room temperature; the solution was transferred to new plates and OD595 was measured using an Eon thermo-regulated spectrophotometer plate reader (BioTek Instruments, USA). The assay was performed three times. The biofilm formation index was calculated as the ratio of OD595 after crystal violet treatment to OD595 before treatment, after deducting from each OD value the average of the bacterium-free medium before and after crystal violet treatment, respectively [38].

### 2.12. Statistical Analysis

The growth kinetic data from WT and mutant strains were compared using a nonparametric, quantile (median) regression given that these data did not present a normal distribution; the dependent variable was the OD600 value and the explanatory variables were “dummy variable” strains, i.e., a categorical variable with a modality for each strain and the WT strains taken as references. The growth rate data were pooled (OD600, T = 23 h; means of triplicates for each measurement, *n* = 165). Regarding biofilm formation, the data from the studied strains was compared using a one-way ANOVA. A Shapiro–Wilk test was used to check normal distribution of the data. The Glycocalyx frequency data was compared using chi2 tests. Statistical analyses were performed using the Stata software package, v. 14.1 (Stata Corp., College Station, TX, USA), GraphPad Prism 6 software and BiostaTGVsoftware; *p*-values of <0.05 were considered significant. 

## 3. Results

### 3.1. Genome Analysis of 12/111phiA

The in silico analysis of the WGS of WT 12/111 confirmed the 60,651 pb prophage genome inserted at a site flanked by *adaA* (upstream) and *ybjI* (downstream) bacterial genes. The 12/111phiA genome exhibits a GC content of 42% and contains 58 ORFs organized in modules and mostly encoded on the reverse strand (Figure 1). The packaging module is followed by three morphogenesis modules dedicated to head, head–tail, and tail, respectively, by the host lysis module. The tail morphogenesis module includes a 1039-codon-long ORF, encoding the tail tape measure protein, highly suggestive of 12/111phiA belonging to the *Siphoviridae* family [39,40,41]. Upstream of the packaging module are the lysogeny and replication modules. The lysogeny module has two peculiarities: first, in silico analysis did not reveal any integrase, but rather a transposase with an integrase core domain; second, CI and the Cro repressor are separated by a 12 kpb long DNA element including a 7035 bp ORF encoding a protein similar to StrP (100% coverage, 100% identity), a protein described in the phiStag1-like prophage associated with the emerging lineage CC17-1A recently described in neonates [19]. Downstream of the replication module is a foreign DNA defence module with a restriction–modification (RM) system. The search for prophage ORFs that could potentially benefit the host revealed a region in the foreign DNA defence module comprising four ORFs, including three which have been associated with cell persistence and biofilm formation [42,43,44]: first, *relB* and *yafQ*, two ORFs in the same direction and separated by three nucleotides, coding for putative RelB antitoxin and YafQ toxin, i.e., a type II TA system [45,46]; and a second couple of ORFs in the same direction and separated by one nucleotide, i.e., one encoding a putative endonuclease and *metK,* a gene encoding a putative S-adenosylmethionine synthetase, an enzyme that catalyzes the conversion of methionine to S-adenosylmethionine (SAM) [16,42]. We compared the prophagic amino acid sequences of RelB, YafQ, and MetK with the available sequences of bacterial functional homolog proteins [45,47] (Appendix A): First, a 26–30% similarity at the amino-acid level (48–66% cover) was found between the sequences of three RelB antitoxins [45] and the 12/111phiA_RelB sequence (Appendix A). Second, 33% identity (44–93% cover) was found between the 12/111phiA_YafQ sequence and those of two RelE toxins [45] (Appendix A). Moreover, the amino acids considered as essential for the functional activity of the YafQ protein were found in the prophage sequence [48,49]. Third, a 44–57% similarity (95–97% cover) was found between the 12/111phiA_MetK sequence and that of 12/111 host MetK, and those of two functional MetK described in *E. coli* [47]. The sequences of 12/111phiA_MetK and *E. coli* MetK differed by a 135-amino-acid truncation*,* but without impacting the domain involved in SAM synthetase activity (Appendix A). Overall, our data argue in favor of five prophagic genes encoding a type II TA system, an endonuclease, and MetK, similar to homolog proteins of bacterial origin and, thus, likely functional for the host, along with a putative StrP-like adhesin.

Using BlastN tools, a search for prophages similar to 12/111phiA identified 10 Javan-like prophages from the genome of *Streptococci* [50], and 12/111phiA-like features in the genomes of 13 GBS strains (Appendix A; Figure 2) and 10 non-GBS *Streptococci*, including *S. urinalis*, *S. constellatus*, *S. canis*, and *S. equi*. In particular, prophage MR1-Z1-201_Phi carried by a ST23 GBS, isolated in the U.K. from a seal, shared 93% identity (78% cover) with 12/111phiA (Figure 2) and yielded three genes identical to the three ORFs *endonuclease*, *yafQ*, and *relB* carried by 12/111phiA (100% identity; 100% cover) [51].

### 3.2. Construction of Mutants

Mutants were constructed to investigate the potential impact of the five identified prophage ORFs of interest on host features. Following mitomycin C exposure of 12/111, we obtained a strain deleted for 12/111phiA (12/111ΔphiA), with similarity to 12/111 (excluding prophage-loss) confirmed by pairwise comparison of their WGSs (Appendix A). Despite multiple trials, we failed to detect 12/111phiA in the broth cultures of 12/111 following exposure to mitomycin C. Using double homologous recombination, we constructed (i) two independent deleted mutants for each of the following five regions: *relB-metK*, *relB-yafQ*, *endonuclease-metK*, *endonuclease* and *metK*, and (ii) in situ and plasmid-complemented mutants for *relB-yafQ*, *endonuclease-metK*, *endonuclease* and *metK* by transformation of 12/111ΔphiA and 12/111Δ*relB-metK* strains (Table 1). Deleted and complemented mutants were verified by sequencing the regions of interest, or using WGS. Regarding *strP*, we failed to obtain mutants and their respective complements, likely due the large size of the ORF (7035 bp). Thus, further experiments were conducted with 35 constructs (Table 1).

### 3.3. Growth Characteristics, Morphology, and Biofilm Formation for 12/111, 12/111ΔphiA, and the Deleted Mutants

WT and 12/111ΔphiA were cultured overnight in TH 1% glucose broth at 37 °C. The macroscopic appearance of the broths differed, with a clear supernatant for 12/111 and a cloudy supernatant for 12/111ΔphiA (Figure 3A). The deposit at the bottom of the bottles also differed: the sediment was flaxy and heterogeneous (i.e., flocculation) for 12/111, whereas it was fine and homogeneous for 12/111ΔphiA. Microscopic examination of the broths revealed additional differences. Using Gram staining and wet mount, 12/111 showed long chains and large chain tangles, while 12/111ΔphiA showed medium length chains and small chain tangles (Figure 3B and Appendix A). Using scanning electron microscopy (SEM), 12/111 showed regular shaped spherical clusters made up of very tight bacteria networks, while 12/111ΔphiA showed irregular shaped clusters made up of a distended bacterial network (Figure 3C). The morphology of the cocci was similar, with homogeneous and regular ovoid cells in both cases. Using transmission electron microscopy (TEM) analysis, the morphology and wall thickness of the cocci were similar for the two strains studied. Glycocalyx-like components were observed with both strains. The proportion of cocci harboring a thick layer of glycocalyx-like features, however, differed significantly: with 12/111, this proportion reached 25% of cells, while it was 5% for 12/111ΔphiA (*p* < 0.001) (Figure 3D) (Table 1). The growth kinetics of 12/111 and 12/111ΔphiA were compared using TH/MRS broth at 37 °C with agitation for 24 h (Figure 4). The growth profiles were similar, except for a higher OD600 nm at stationary phase for 12/111 (*p* < 0.001) (Table 1). The microbial counts of 12/111 and 12/111ΔphiA were however similar when studied after 23 h incubation (Appendix A). 12/111 and 12/111ΔphiA were studied further for their ability to form biofilm, and the biofilm formation index (BFI) was significantly higher for 12/111 than for 12/111ΔphiA (*p* < 0.0001) (Table 1; Figure 5). Overall, 12/111 and 12/111ΔphiA differed significantly regarding (i) the presence or absence of flocculation, (ii) the length of the chains and the size of chain tangles, (iii) the shape of the cocci clusters, (iv) the prevalence of cocci with a thick layer of glycocalyx, (v) growth kinetic characteristics (maximum OD600 during the stationary phase), and (vi) the ability to form biofilm.

The 12 remaining deleted mutants were phenotypically similar to 12/111ΔphiA with no flocculation, medium-length chains and small chain tangles, similar growth kinetic characteristics, and low ability to form biofilm (Table 1).

### 3.4. Restoration of the Phenotypic Traits of the Complemented Mutants

To rule out the eventuality of the phenotypic variations observed with the deleted mutants being the consequence of secondary mutations, complemented mutants were constructed and studied for phenotype restoration.

#### 3.4.1. In Situ Complemented Mutants

One in situ complemented mutant of the *relB-metK* region, i.e., 12/111Δ*relB-metK::relB-metK* n°2, totally restored the phenotypes studied: (i) flocculation, (ii) long chains and large chain tangles, (iii) high prevalence (21.5%) of cocci with a thick layer of glycocalyx, (iv) growth characteristic of 12/111, and (v) high ability to form biofilm (Table 1; Figure 4 and Figure 5). The remaining in situ complemented mutants did not restore any of the studied phenotypes, suggesting the onset of secondary mutations in the course of mutant’ construction, potentially preventing phenotype restoration. 

#### 3.4.2. Plasmid Complemented Mutants

12/111, 12/111ΔphiA and the mutants were first transformed with empty pTCV-P_Tet_ to assess plasmid impact on the studied phenotypes. The comparison of each strain with its respective plasmid control strain revealed no plasmid effect for all but the 12/111 pTCV-P_Tet_ strain (Table 1; Appendix A). In light of this, we used 12/111 as a technical positive control, and plasmid-complemented mutants were compared to mutant strains carrying pTCV-P_Tet_. Among the 10 plasmid-complemented mutants, 4 partially restored the studied phenotypes (i.e., 12/111ΔphiA pTCV-p_Tet_::*endonuclease-metK,* 12/111ΔphiApTCV-p_Tet_::*metK,* 12/111Δ*relB-metK* pTCV-p_Tet_::*endonuclease-metK and* 12/111Δ*relB-metK* pTCV-p_Tet_::*metK*) (Table 1): in all cases, they produced flocculation, and long chains and large chain tangles (Figure 3A,B). The ability to form biofilm and the growth kinetic characteristics were not restored (Table 1) (Appendix A), nor was the high prevalence of cocci with a thick layer of glycocalyx, except for 12/111Δ*relB-metK* pTCV-p_Tet_::*metK and* 12/111ΔphiA pTCV-p_Tet_::*metK* (*p* < 0.005). SEM examination of the strains carrying pTCV-p_Tet_::*endonuclease-metK* or pTCV-p_Tet_::*metK* showed frequent irregular, deformed, acorn-shaped cocci, highly suggestive of abnormal septation in these strains (Figure 6). The remaining six plasmid-complemented mutants did not restore the analyzed phenotypes, again suggesting a secondary mutation effect.

### 3.5. Search for Secondary Mutations in Deleted and In Situ Complemented Mutants

To investigate the non-restoration of the studied phenotypes in a proportion of our complemented mutants, two mutants and their complements were fully sequenced. The WGS were circularized and aligned ether with the WT strain (12/111) or with complement strains, and two-by-two genome comparisons were performed to detect mutations (Appendix A). No non-synonymous mutation was found between the mutants and the WT 12/111 strain, and no mutation was found common to the mutants and absent from the WT. Genome comparison of 12/111Δ*relB-metK* n°2 and its complement 12/111Δ*relB-metK::relB-metK* n°2 (i.e., that restoring the phenotypes) revealed six mutations, three of which in an intergenic region and three synonymous. By contrast, for the pair of remaining mutant and complement (i.e., that not restoring the phenotypes), genome comparisons revealed mutations in intergenic regions, but also, non-synonymous mutations in the complement: one in an ORF encoding a putative Xylulose Kinase, and the second in an ORF encoding a putative Vex1 protein component of an ABC transporter ensuring export of Pep^27^, a peptide involved in growth inhibition, cell death and cell wall autolysis [52,53]. Taking into account the genes affected, it seems plausible that the secondary mutations identified in the genome of the complemented mutants may have hindered or disturbed restoration of phenotype complementation.

Overall, the loss of the WT 12/111 phenotypes studied in *relB-metK* mutants and their restoration, complete in the in situ 12/111Δ*relB-metK::relB-metK* n°2 complemented mutant, and partial in the plasmid complemented mutants harboring pTCV-p_Tet_::*endonuclease-metK* or pTCV-p_Tet_::*metK,* demonstrate a role of the *relB-metK* region, and particularly *metK*, in the clumping of the cocci in chains, cell autoaggregation and in the formation of glycocalyx and biofilm.

## 4. Discussion

Over the past two decades, group-A prophages have been identified from the genome of GBS and other Streptococci isolated from infected animals (i.e., cattle, swine, fish and seal), and epidemiological studies conducted worldwide have established that CC17 GBS carrying group-A prophages in their genome are increasingly responsible for neonatal diseases [17,18]. Our study, designed to investigate the benefit of prophage A lysogeny in the host, provides important new data. To the best of our knowledge, this is the first time that a functional impact of prophage ORFs has been demonstrated on a GBS host phenotype by using deleted and complemented mutants.

We first conducted an in-depth study of the group-A prophage 12/111phiA. By in silico analysis, we confirmed that 12/111phiA inserted close to host genes is involved in adaptation to acid stress and virulence in *M. tuberculosis, S. pneumoniae*, and *E. coli* (i.e., *adaA* and *murB-potABCD*) [16,18,54,55,56]. The prophage genome presented major peculiarities. First, it was devoid of transcriptional regulation genes [24]. Second, the lysogeny module appeared with an unusual structure, with the ORF encoding CI and Cro repressor very far apart due to a 12 kbp long DNA fragment between them; furthermore, concordant with the findings of Crestani et al., no integrase was found by in silico analysis [57]. Third, it had a putative RM system in the foreign DNA defence module [58], along with a putative TA system [45] close to the putative RM system [59,60,61], suggesting a prophage maintenance strategy. Fourthly, we found five ORFs of interest for the host, encoding (i) a StrP-like adhesin [19], (ii) a type II TA system RelB/YafQ associated with bacterial persistence under stress conditions and biofilm formation [43,59,62,63], (iii) an endonuclease, and (iv) a SAM synthetase MetK (i.e., an enzyme catalysing the synthesis of SAM [42,64]), a key component in the methylation of DNA, RNA and proteins, in the synthesis of polyamines [64], and in the production of the signaling molecule auto-inducer 2 (AI-2) associated with autoaggregation and biofilm formation [65,66,67,68].

Prophage excision was obtained following mitomycin C exposure of the lysogen. Despite multiple trials, however, we did not detect 12/111phiA in the broth cultures of 12/111 following exposure to mitomycin C. These findings suggest a difficult-to-induce prophage. Defective prophages have been shown as drivers of evolutionary phenotype and fitness changes in different species, and they may confer an advantage to lysogens in challenging environments by contributing to acid, antibiotic, and oxidative stress resistance [69,70,71]. The absence of prophage transcriptional regulation genes in the genome of 12/111phiA, along with the unusual characteristics of its lysogeny module, may explain the difficulties encountered for induction. These data, in addition with the conservation of prophage elements that may benefit the host, are elements arguing in favor of a notable interest for the host to maintain 12/111phiA in its genome, a mechanism named prophage domestication by the host [72].

The functional impact of 12/111phiA and the identified ORFs of interest on 12/111 was investigated using a prophage-free strain and a series of deleted mutants and their respective complements. The phenotypic differences observed between the lysogenic strain and the mutants regarding growth characteristics, cell morphology, and ability to form biofilm argued in favor of a contribution from the studied prophage genes to autoaggregation and biofilm formation in the lysogenic strain. Indeed, the presence of the prophage features in the host genome was associated with (1) flocculation in liquid cultures, corresponding to microbial aggregates [73]; (2) long cell chains, a characteristic associated with autoaggregation shown as a key determinant of biofilm formation [74,75,76,77]; (3) a particular ability to produce glycocalyx, a polysaccharide and protein film surrounding bacterial cells and forming part of the biofilm [78]; and (4) a high ability to form biofilm. By contrast, deletion of the *relB-metK* region or any studied part of this region resulted in the loss of flocculation in liquid cultures, shorter chains, and a significantly decreased prevalence of cells surrounded by glycocalyx and biofilm formation. Moreover, a full restoration was obtained for one in situ complemented mutant of the *relB-metK* region, a fundamental result establishing that the prophage *relB-metK* region influences host cell morphology and glycocalyx and biofilm formation. These data are concordant with the YafQ type II TA system having been shown to influence biofilm formation [79]. Partial restoration was also achieved with four plasmid-complemented mutants harbouring pTCV-pTet::*endonuclease-metK* and pTCV-pTet::*metK* with the flocculation recurring and the long chains for all the mutants, and the ability to produce glycocalyx with pTCV-pTet::*metK*. Remarkably, the four plasmid-complemented mutants displayed stronger autoaggregation than that observed with 12/111, along with frequent cell heterogeneity and abnormal septation. In *S. suis,* long chains with abnormal septation have been associated with unstable peptidoglycan due to the deletion of *potA,* a major part of the potABCD polyamine transporter [80]. SAM is a key determinant of the synthesis of AI-2 and of polyamines [64,65], i.e., molecules which play a role in peptidoglycan synthesis in *Streptococci* [80,81] and in biofilm formation [82,83]. Thus, we suggest that (1) the stronger autoaggregation and higher cell heterogeneity with abnormal septation observed in our study with the plasmid-complemented strains may have resulted from increased synthesis of AI-2 and polyamines due to the overexpression of *metK* carried on the plasmids, and (2) *metK* may play a role in quorum sensing and peptidoglycan synthesis.

A particularity of our work was to explore why total restoration of the phenotypic traits of 12/111 was not achieved for several complemented mutants. Two-by-two WGS comparisons revealed no non-synonymous secondary mutations in the mutants or in the complemented mutant which fully restored the 12/111 phenotypes, and by contrast, in the complemented mutants which failed phenotype restoration, it revealed non-synonymous mutations in ORFs involved in growth inhibition, biofilm formation, cell death, or cell wall autolysis [52,53]. The presence of mutations in such ORFs in the genome of the complemented strains that did not restore the expected phenotypes and the absence of such mutations in the genome of the complemented strain for which restoration was fully achieved provide a plausible explanation as to the non-restoration of the WT phenotypes.

In summary, the WT 12/111 strain carries a domesticated prophage, 12/111phiA, inserted close to bacterial genes involved in adaptation to stress, and it harbors in its genome five ORFs encoding a putative adhesin StrP and bacterial homolog proteins that may influence autoaggregation and biofilm formation of the lysogenic host. Autoaggregation may be triggered under conditions of environmental stress and is defined by bacteria of the same strain forming multicellular clumps, within which they are protected from environmental stresses (i.e., toxins, antibiotics, predation, or lack of nutrients) and host responses [66,84,85]. Autoaggregation is also one of the first steps in biofilm formation [86]. Biofilm is a bacterial persistence form consisting of a community of sessile bacteria, encompassed in an extracellular matrix made up of extracellular polymeric substances, which may adhere to and colonize abiotic surfaces or organic tissues [87,88]. Biofilm has been shown to improve the colonization of challenging environments by increasing antimicrobial tolerance [89], promoting resistance to bacteriocin activity [90] and, to unfavorable conditions such as acid stress [91] or an osmotic medium [92], and protecting against the host immune system [93]. Moreover, in the biofilm matrix, some enzymes are involved in the degradation of extracellular polymeric substances, which deliver energy and carbon sources to bacterial biofilm cells, especially in a nutrient-restricted environment [94]. Overall, autoaggregation and biofilm formation are factors favoring the colonization of ecological niches and bacterial persistence facing conditions of stress in numerous species, including *Streptococci* [86,95,96]. Colonization of the vaginal tract by GBS is a major risk factor for neonatal infection [97]. In this ecological niche, by producing H_2_O_2_, lactic acid, bacteriocins, and biosurfactants and by adhesion competition, *Lactobacilli* mediate vaginal homeostasis and make it difficult for other bacteria to establish themselves [98,99]. By promoting autoaggregation and biofilm formation in the lysogen, we suggest that 12/111phiA may confer an advantage on GBS strains for adaptation to or colonization of the maternal vaginal tract by increasing GBS persistence in this stressed environment; it consequently may increase the risk of neonatal infection in colonized parturients.

## 5. Conclusions

Our study, designed to investigate whether acquisition of group-A prophages should favor adaptation in humans, revealed a functional impact of lysogeny on the host. These data argue for a significant role of lysogeny in increasing ability of GBS to cause infections in humans and suggest that the acquisition of these prophages may have played a role in the epidemiological changes currently observed in GBS species, including the trend towards an increase in neonatal infections. Additional investigations should be performed to explore (i) the impact of the insertion of 12/111phiA on the function of the adjacent bacterial genes (i.e., *ybjI*, *adaA*, *clcA* and *murB-potABCD* operon), (ii) the level of SAM and AI-2 in cells differing in *metK* gene [100,101,102], and (iii) the impact of 12/111phiA using an in vivo model on GBS antimicrobial tolerance and pathogenicity.

## Figures and Tables

**Figure 1 microorganisms-09-01112-f001:**
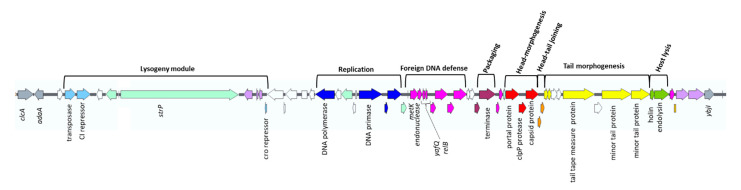
Prophage 12/111phiA genome organization. Bacterial genes adjacent to phage attachment are gray; ORFs belonging to the same module are of a similar color; hypothetical proteins are white; transposases are purple; ORFs annotated with any modular function are light green. The putative function of the ORFs is shown.

**Figure 2 microorganisms-09-01112-f002:**
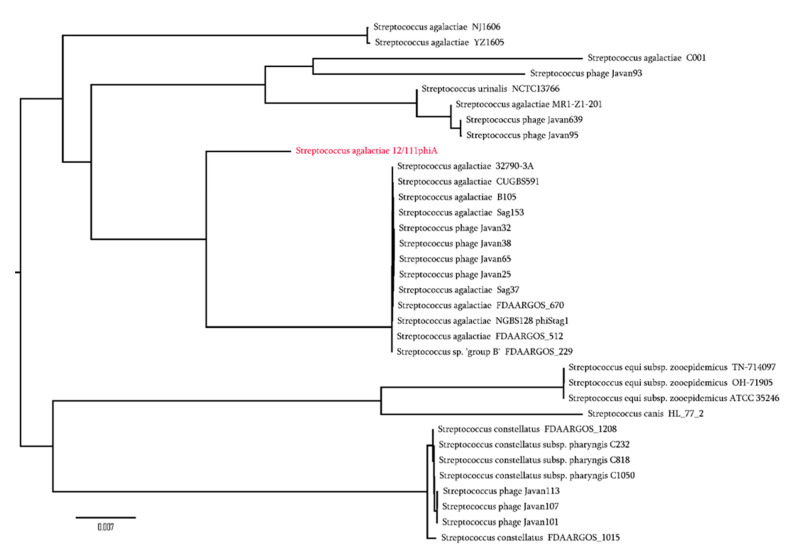
Phylogenic tree of 12/111phiA prophage with the 33 best hit sequences from the NCBI database with % identity ≥90% and % cov alignment ≥60%, including prophagic sequences highly similar to 12/111phiA. All these hit sequences with their % identity and % coverage alignment are presented in Appendix A.

**Figure 3 microorganisms-09-01112-f003:**
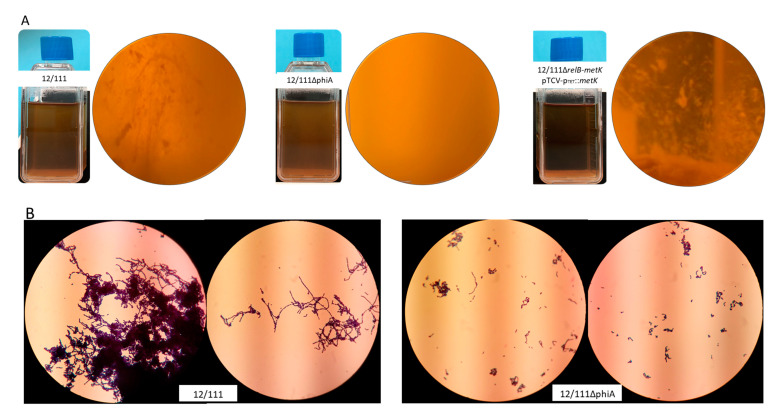
Growth characteristics of 12/111, 12/111ΔphiA and 12/111Δ*relB-metK* pTCVp_TET_::*metK*: (**A**) macroscopic appearance of TH 1% glucose broth (flask) and culture sediment (round section); (**B**) bacterial examination by Gram staining (×100 objective); (**C**) bacterial cluster examination by SEM; (**D**) glycocalyx observation by TEM, where arrows indicate glycocalyx-like structures.

**Figure 4 microorganisms-09-01112-f004:**
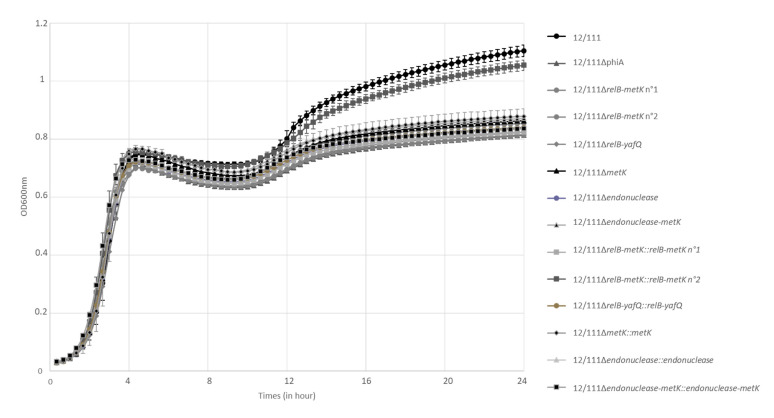
Growth kinetics of 12/111, 12/111ΔphiA, deleted and in situ complemented mutants. Growth was performed in TH/MRS medium at 37 °C with agitation. Each point represents the mean OD600 measurement from three independent experiments.

**Figure 5 microorganisms-09-01112-f005:**
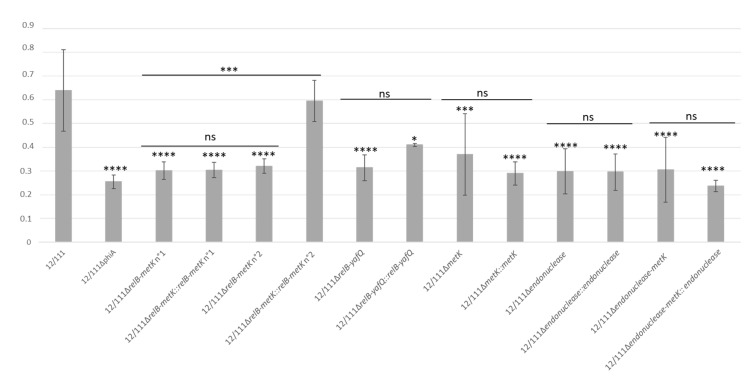
Biofilm formation of 12/111, deleted mutants and in situ complemented mutants in TH 1% glucose. * *p* < 0.05; *** *p* < 0.001; **** *p* < 0.0001; ns = not significant. Data represent the mean biofilm formation index (BFI) from three independent experiments.

**Figure 6 microorganisms-09-01112-f006:**
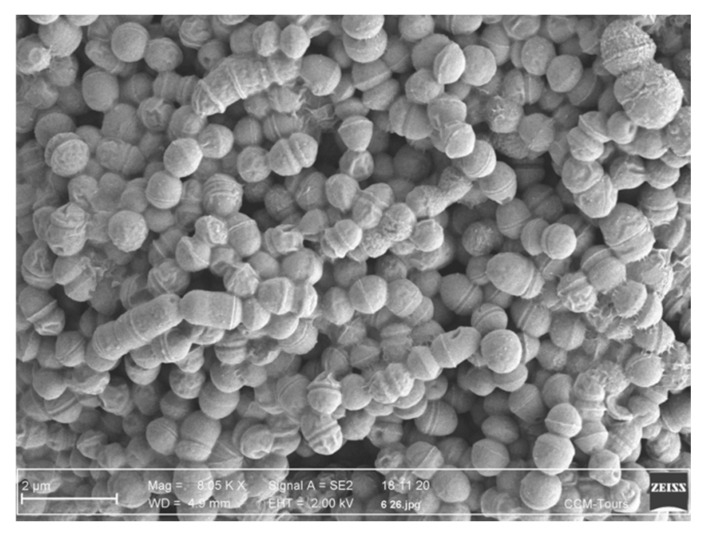
12/111Δ*relB-metK* pTCVpTET::*metK* cocci observation by scanning electron microscopy.

**Table 1 microorganisms-09-01112-t001:** Strain behaviour according to tested phenotypes. Supernatant C clear, T turbid; G1 corresponding to long chains with large tangles, G2 medium chains with small tangles. Maximum growth is compared to wildtype strains * as a percent. Biofilm formation index results are expressed compared to wildtype strains* as a percent. * 12/111 for plasmid-free strains and 12/111 pTCV-P_Tet_ n°1 for strains carrying plasmid; SEM analysis: R = Regular and tight spherical structures, I = Irregular and distended structure. R = Homogeneous and regular ovoid cells, I = Irregular, deformed cocci. TEM analysis: SD = standard deviation.

Strains	Culture Aspect	Gram Stanning	Maximal Growth	Biofilm Formation Index	SEM		TEM	
					Cluster Appearance	Cocci Morphology	Wall Thickness (in nm)	Glycocalyx Frequency
12/111	C	G1	100	100	R	R	14.0 (SD = 4.41)	23%
12/111∆phiA	T	G2	73.8 (*p* < 0.0001)	39.8 (*p* < 0.0001)	I	R	14.3 (SD = 1.88)	5%
Deleted mutants								
12/111∆*relB-metK n°1*	T	G2	73.9 (*p* < 0.0001)	47.2 (*p* < 0.0001)	-	-	-	-
12/111∆*relB-metK n°2*	T	G2	77.7 (*p* < 0.0001)	50.1 (*p* < 0.0001)	R/I	R	13.4 (SD = 1.84)	3.5%
12/111∆*relB-yafQ*	T	G2	74.8 (*p* < 0.0001)	49.0 (*p* < 0.0001)	-	-	-	-
12/111∆*metK*	T	G2	78.4 (*p* < 0.0001)	57.9 (*p* < 0.001)	-	-	-	-
12/111∆*endonuclease*	T	G2	76.1 (*p* < 0.0001)	46.6 (*p* < 0.0001)	-	-	-	-
12/111∆*endonuclease-metK*	T	G2	79.2 (*p* < 0.0001)	47.6 (*p* < 0.0001)	-	-	-	-
in situ complemented mutants								
12/111Δ*relB-metK::relB-metK n°1*	T	G2	74.9 (*p* < 0.0001)	47.6 (*p* < 0.0001)	-	-	-	-
12/111Δ*relB-metK::relB-metK n°2*	C	G1	95.5 (ns)	93.0 (ns)	R/I	R	13.5 (SD = 1.20)	21.5%
12/111∆*relB-yafQ::relB-yafQ*	T	G2	76.2 (*p* < 0.0001)	64.1 (*p* < 0.05)	-	-	-	-
12/111∆*metK::metK*	T	G2	79.8 (*p* < 0.0001)	45.3 (*p* < 0.0001)	-	-	-	-
12/111∆*endonuclease::endonuclease*	T	G2	73.0 (*p* < 0.0001)	46.2 (*p* < 0.0001)	-	-	-	-
12/111∆*endonuclease-metK::endonuclease-metK*	T	G2	76.2 (*p* < 0.0001)	37.2 (*p* < 0.0001)	-	-	-	-
12/111 pTCV-P_Tet_ n°1	T	G2	100	100	-	-	-	-
12/111 pTCV-P_Tet_ n°2	T	G2	100.2 (ns)	87.1 (ns)	-	-	-	-
12/111∆phiA pTCV-P_Tet_	T	G2	99.4 (ns)	61.5 (*p* < 0.01)	R/I	R	13.4 (SD = 4.06)	2%
12/111∆phiA plasmidic complemented mutants								
12/111∆phiA *pTCV-P_Tet_::relB-yafQ*	T	G2	96.5 (ns)	65.8 (*p* < 0.05)	-	-	-	-
12/111∆phiA *pTCV-P_Tet_::metK*	C	G1	97.9 (ns)	66.0 (*p* < 0.05)	R/I	I	12.3 (SD = 3.07)	6%
12/111∆phiA *pTCV-P_Tet_::endonuclease*	T	G2	93.3 (ns)	58.3 (*p* < 0.001)	-	-	-	-
12/111∆phiA *pTCV-P_Tet_::endonuclease-metK*	C	G1	91.1 (ns)	54.0 (*p* < 0.0001)	R/I	I	11.5 (SD = 1.53)	3%
12/111Δ*relB-metK pTCV-P_Tet_*	T	G2	95.6 (ns)	79.5 (ns)	R/I	R	13.5(SD = 3.86)	4%
12/111∆*relB-metK* plasmidic complemented mutants								
12/111∆*relB-metK pTCV-P_Tet_::relB-yafQ*	T	G2	100.0 (ns)	62.8 (*p* < 0.01)	-	-	-	-
12/111∆*relB-metK pTCV-P_Tet_::metK*	C	G1	91.2 (ns)	51.3 (*p* < 0.0001)	R/I	I	10.9 (SD = 0.93)	12.5%
12/111∆*relB-metK pTCV-P_Tet_::endonuclease*	T	G2	99.2 (ns)	61.3 (*p* < 0.01)	-	-	-	-
12/111∆*relB-metK pTCV-P_Tet_::endonuclease-metK*	C	G1	102.1(ns)	71.4 (ns)	R/I	I	11.3 (SD = 1.93)	4%

## Data Availability

All DNA sequences obtained during the study were deposited in the GenBank database at https://www.ncbi.nlm.nih.gov/genbank/ (accessed on 17 May 2021).

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
