# Peer review of "12/111phiA Prophage Domestication Is Associated with Autoaggregation and Increased Ability to Produce Biofilm in Streptococcus agalactiae"

_microorganisms, 2021, doi:10.3390/microorganisms9061112_

Round 1

Reviewer 1 Report

The manuscript by Renard et al describes the analysis of the effect of prophage within a clinical isolate of GBS that was highly virulent.  Using molecular techniques, they cured the strain of its prophage and compared phenotypes between the lysogen and prophage cured strain to determine if the phage may be responsible for any of the differences.  There was a great amount of work done here with the construction of 43 different strains. However, the only new info that came from this paper was that the prophage caused these phenotypes, which could have been shown without making all of the mutants.  In addition, there appears to be a lack of understanding of basic bacteriophage biology that makes some of the other assumptions by the authors somewhat questionable. 

            First, the authors state that there is no CI repressor, but they show on the phage genome both a cro repressor and an immunity repressor.  The immunity repressor IS the CI repressor. So, this does not make it a defective prophage.  Second, it is not clear how they were able to obtain a phage-cured strain in the absence of an integrase.  Both excise AND integrase are needed for a prophage to excise from the chromosome. Therefore, there must be an integrase present. Also, if the prophage can excise from the chromosome, then it is not a defective prophage.  I would like to hear an explanation as to why they felt they were able to get a phage cured strain if it is considered to be defective. Third, one of the prophage genes they investigate is the clpP-like protease that is found in the structural region of the prophage, right next to the portal protein.  Although HHPRED will identify it as a ClpP protease, in bacteriophage it is called a “capsid maturation protease” and is always right in between the portal protein and next to a capsid protein, as it is in this phage genome.  It is required for degradation of parts of the prohead, needed for assembly, but needs to be removed for maturation of the prohead to the head of the virus particle. The presence of this “clpP-like protease” in this region of the prophage genome is very typical for a bacteriophage and not indicative of a protein of bacterial origin or functional for the host – this is needed for the phage.

            One question would be whether they were able to identify the att sites of the phage, to clearly define where the phage inserted into the chromosome.  Was attB in the bacteria recreated when they lost the prophage?  If so, did they look for phage particles in the supernatant? If so, they could just remake the lysogen by infecting with the phage and showing that the phenotypes are restored.

            For the Gram stains in Fig 3B, it would have been much better to see a more diluted cultre for the 12/111 so that we could see individual chains. There were clearly fewer bacteria in the phage-cured strain for the Gram stain. A micrograph of a wet mount on a slide before Gram staining would show the differences they want to show. We do this often when looking at chain length differences.

            There are also multiple typos and grammatical problems with the writing, which I am sure has to do with English being a second language.  For example: Table 1 legend – T should be for “turbid” not “trouble”, “no GBS” should be “non GBS”, etc.

Reviewer 2 Report

In my opinion this is a very good manuscript . I reccommend its publication

Just two suggestions:

1.- L 43 “Temperate bacteriophages (prophages) may integrate their genome into the bacterial  chromosome and confer evolutionary or adaptive advantage to the host [8]…Move prophages as follows: “Temperate bacteriophages (prophages) may integrate their genome into the bacterial chromosome (prophages) and confer evolutionary or adaptive advantage to the host [8]

2.- P 18 L 83  Streptococci should not be italics.

Author Response

Rewiever 2

Our responses

In my opinion this is a very good manuscript. I reccommend its publication

Just two suggestions:

1.- L 43 “Temperate bacteriophages (prophages) may integrate their genome into the bacterial  chromosome and confer evolutionary or adaptive advantage to the host [8]…Move prophages as follows: “Temperate bacteriophages (prophages) may integrate their genome into the bacterial chromosome (prophages) and confer evolutionary or adaptive advantage to the host [8]

Done (lines 42-44).

2.- P 18 L 83  Streptococci should not be italics.

Done (lines 457).

Reviewer 3 Report

The manuscript investigates impact of prophage phiA on properties of its GBS host strain by construction of several deleted and complementation mutants. The authors have shown that phiA could be responsible for cell clumping and biofilm formation and they tried to found phage genes which confer observed properties. However, several inconsistent results were obtained in the study and their explanations are sometimes insufficient.

The manuscript is also quite long with many considerable details which make reading difficult. A lot of mutants were prepared, but their biological properties differed from the expectations of the authors and these results were not satisfactory explained. I recommend to shorten the text to the most important and relevant findings. There are many inaccuracies in the text and I recommend strict editing before publication.

It should be also appropriate to study phenotypes of mutants in more detail. Especially would be nice to detect level SAM (or AI-2) in cell differing in metK gene.

Finally I recommended the manuscript for major revision.

p. 7, l. 315: The 12/111phiA phage is drawn in reverse orientation on Fig 1. Please, change the direction to give sense orientation to the majority of genes. Change “genes binding prophages” to “genes adjacent to phage attachment”

p. 7, l. 320: Change “features into the genomes” to “features in the genomes”

p. 7, l. 321: Change “10 no GBS” to “10 non GBS”

p. 8, l. 338: Omit “strP”

p.13: Figure 3: Change legend to “(A) macroscopic appearance of TH 1% glucose broth (flask) and culture sediment (round section)“ I is not necessary to include two same figures of SEM for each strain. There is not provided the same magnification of TEM figures for all strains therefore the comparison is not possible. The glycocalyx should be marked in the figure for the better clarity.

p.14: Figure 4: The data are not easy to read. It is not clear which strains grew faster than others. I suggest to omit some strains from the picture. Why mixed TH/MRS broth was used for the experiments?

p. 15, l. 14: Change “totally restored all but one phenotype studied with” to “restored studied phenotypes including”

p. 15, l. 19: Include “(Fig. 6)”

p. 15-16, l. 41-64: Secondary mutations should be preferentially compared to the wild-type strain (12/111) as these differences could be the cause of phenotype changes in complementation strains but also in deletion mutants. Quite similar phenotypes were observed in all mutants (decreased glycocalyx production and lowered biofilm). It could be possible that this was caused by a secondary mutation common to all mutants.

p. 16, l. 65: Figure 7 is not informative and should be removed from the manuscript.

p. 17, l. 70: Table 1: It is not necessary to present results of biological replicates of mutants if replicates showed similar behaviour. It will make simple the understanding. The same is true for Figure 4 and 5.

p.18, l. 82: Change “the genome of GBS and no GBS Streptococci isolated from infected animals” to “the genome of GBS of human origin and not in GBS Streptococci isolated from infected animals”

p. 18, l. 96: If the phage 12/111phiA was defective, it would not to be possible to prepare cured strain with mitomycin C induction. Was the presence of phiA phage after mitomycin C induction tested in culture?

p. 21, l. 266: Change ”Jamrozy, D.; de Goffau, M.C.; Bijlsma, M.W.; van de Beek, D.; Kuijpers, T.W.; Parkhill, J.; van der Ende, A.; Bentley, S.D. Temporal Population Structure of Invasive Group B Streptococcus during a Period of Rising Disease Incidence Shows Expansion of a CC17 Clone; Genomics, 2018;“ to ” Jamrozy et al. Increasing incidence of group B streptococcus neonatal infections in the Netherlands is associated with clonal expansion of CC17 and CC23, Scientific Reports 2020”

Author Response

Rewiever 3

Our responses

The manuscript investigates impact of prophage phiA on properties of its GBS host strain by construction of several deleted and complementation mutants.

The authors have shown that phiA could be responsible for cell clumping and biofilm formation and they tried to found phage genes which confer observed properties.

However, several inconsistent results were obtained in the study and their explanations are sometimes insufficient.

We have modified the text taking into account all your requests.

The manuscript is also quite long with many considerable details which make reading difficult.

I recommend to shorten the text to the most important and relevant findings.

The revised manuscript has been simplified and shortened. As requested by another reviewer, we have removed the study regarding clpP.

A lot of mutants were prepared,

but their biological properties differed from the expectations of the authors and these results were not satisfactory explained.

We have first constructed -         a prophagefree strain, -         for each ORF of interest, 2 independent deleled mutants.

All the deleted mutants have lost the phenotypes of the WT, suggesting a role of the studied prophage ORFs in the phenotypes of the WT strain.

With the aim of increasing robustness of these first results, we have then constructed for each of the deleted mutants, independant complements.

We did present all the results obtained :

-         those of complements restoring the WT phenotypes, establishing with force the role of the considered ORFs in the phenotypes studied,

-         and those of complements restoring partially or not restoring the phenotypes.

Regarding these latter unexpected results, we have search and found secondary mutations that could explain the non complementation.

There are many inaccuracies in the text and I recommend strict editing before publication.

The text has been proofread and corrected by a specialized company (A.D.T. International - L'Agence de Traduction, 95230 Soisy - sous – Montmorency).

It should be also appropriate to study phenotypes of mutants in more detail. Especially would be nice to detect level SAM (or AI-2) in cell differing in metK gene.

We agree with your remark. This is indeed a point we will study. We have added this perspective into the discussion part (lines 580-583).

Finally I recommended the manuscript for major revision.

p. 7, l. 315: The 12/111phiA phage is drawn in reverse orientation on Fig 1. Please, change the direction to give sense orientation to the majority of genes. 

The figure has been modified as requested.

Change “genes binding prophages” to “genes adjacent to phage attachment”

Done (line 311).

p. 7, l. 320: Change “features into the genomes” to “features in the genomes”

Done (line 316).

p. 7, l. 321: Change “10 no GBS” to “10 non GBS”

Done (line 319).

p. 8, l. 338: Omit “strP”

Done.

p.13: Figure 3: Change legend to “(A) macroscopic appearance of TH 1% glucose broth (flask) and culture sediment (round section)“ 

Done.

I is not necessary to include two same figures of SEM for each strain. There is not provided the same magnification of TEM figures for all strains therefore the comparison is not possible. The glycocalyx should be marked in the figure for the better clarity.

We agree with your remark. We modified the selected TEM figures. Glycocalyx is now marked in the figure.

p.14: Figure 4: The data are not easy to read. It is not clear which strains grew faster than others. I suggest to omit some strains from the picture.

The figure has been simplified as requested.

Why mixed TH/MRS broth was used for the experiments?

In our global project, we aimed to study the competition between GBS and Lactobacilli. For this purpose, we search for a broth where GBS and Lactobacilli are able to grow.

p. 15, l. 14: Change “totally restored all but one phenotype studied with” to “restored studied phenotypes including”

Done (line 393).

p. 15, l. 19: Include “(Fig. 6)”

Figure 6 is illustrating the deformed cocci observed with the plasmid complements (lines 415-417).

p. 15-16, l. 41-64: Secondary mutations should be preferentially compared to the wild-type strain (12/111) as these differences could be the cause of phenotype changes in complementation strains but also in deletion mutants.

Quite similar phenotypes were observed in all mutants (decreased glycocalyx production and lowered biofilm). It could be possible that this was caused by a secondary mutation common to all mutants.

The deleted mutants have been compared tu the WT strain 12/111. This comparison did not reveal any non-synonymous mutation betwwen the WT strain and each deleted mutant.

Our findings allow us to refute this hypothesis :

-         the analysis of the mutants did not reveal any such secondary mutation, common to all the mutants;

-         the complements restoring the phenotypes did not either present any mutation.

This point is clarified in the revised version (lines 426-429).

 p. 16, l. 65: Figure 7 is not informative and should be removed from the manuscript.

The figure has been removed from the revised version.

p. 17, l. 70: Table 1: It is not necessary to present results of biological replicates of mutants if replicates showed similar behaviour. It will make simple the understanding.

The same is true for Figure 4 and 5.

As requested, replicates showing similar behavior have been removed fom table 1 and from the figures 4 and 5.

p.18, l. 82: Change “the genome of GBS and no GBS Streptococci isolated from infected animals” to “the genome of GBS of human origin and not in GBS Streptococci isolated from infected animals”

We agree this phrase was confusing. The text has been modified (line 457).

p. 18, l. 96: If the phage 12/111phiA was defective, it would not to be possible to prepare cured strain with mitomycin C induction. Was the presence of phiA phage after mitomycin C induction tested in culture?

The phage-free strain was obtained following exposure to mitomycin C of WT 12/111 with a protocol similar to the one we usually perform to induce lysogenic strains. After exposure of WT12/111 to MMC, the broth were centrifugated and the pellet plated on blood agar. After 48h incubation at 37°C, all the colonies were tested using phiA-specific primers (lines 96-104). Most of the tested isolates were not cured. The loss of the phage may be an exceptional event.

We agree with your remark : the prophage was able to excise from the GBS chromosome following exposure to mitomycin C. Using WGS, we have checked the clonal relationship between the lysogen and the cured strain.

Therefore, we agree with 12/111phiA may not be considered defective.

Despite a good experience of GBS and S. aureus prophage induction in our lab, we did not succeed with 12/111phiA induction. We have carried out numerous tests for this purpose with WT 12/111, but we did not obtain detectable phage production (lines 104-108 in the methods section, lines 332-333 in the results section, lines 484-487 in the discussion section).

We only obtained a group-A prophage induction with GBS strains carrying two or three prophages into their genomes (including one belonging to group-A); and in these cases, several phages (including group-A phages) were produced, mixed in the lysates. Phage multiplication and isolation assays have been performed but remained unsuccessful. Our experiments suggest that 12/111phiA may need a help provided by another prophage, to be induced easily.

We suggest that the absence of transcription regulation genes and the presence of an unusual lysogeny module, may play a role with the difficult induction of 12/111phiA.

We have removed the notion of defective prophage from the revised version (lines 490-492 in the discussion section).

p. 21, l. 266: Change ”Jamrozy, D.; de Goffau, M.C.; Bijlsma, M.W.; van de Beek, D.; Kuijpers, T.W.; Parkhill, J.; van der Ende, A.; Bentley, S.D. Temporal Population Structure of Invasive Group B Streptococcus during a Period of Rising Disease Incidence Shows Expansion of a CC17 Clone; Genomics, 2018;“ to ” Jamrozy et al. Increasing incidence of group B streptococcus neonatal infections in the Netherlands is associated with clonal expansion of CC17 and CC23, Scientific Reports 2020

Done.

Round 2

Reviewer 1 Report

I have reviewed the revised manuscript and agree with the changes that were made.  I believe that it has been significantly improved.

Author Response

The revised manuscript has been reviewed by A.D.T. International (traduction).

Reviewer 3 Report

The authors incorporated all my particular comments and suggestions into the manuscript. I appreciate that this substantially increased its value. The text is still too long and sometimes not well written, more editing and shortening should be valuable. However, I recommend it for publication after minor revision.

Author Response

The revised manuscript has been reviewed by A.D.T. International (traduction).

The text has been shortened.